# Minibatches can make neural network training repeatable for clinical applications

Alex Breuer, Larissa V. Furtado, Brent A. Orr

Department of Pathology
St Jude Children's Research Hospital
Memhpis, TN, 38105
Email: {Alex.Breuer,Larissa.Furtado,Brent.Orr}@stjude.org

*Abstract*—To qualify for clinical use, an AI classifier must first satisfy a set of requirements to ensure its safety and efficacy. We concentrate on *longitudinal consistency* over its deployment lifecycle, which may be summarized as: different builds or versions of the classifier must not conclusively produce discrepant results. This requirement is important when a classifier is used for serial monitoring or for stratification for a clinical trial. Our main contribution is an analysis that explains how round-off error can cause training repeatability issues bad enough to create difficulties for longitudinal consistency, and how these training issues may be remediated by using minibatches when training. This analysis is based on a simple neural network that fails to satisfy longitudinal consistency when trained on different hardware due to round-off error that is not controlled by `PyTorch`'s deterministic training mode, even when all software, training hyperparameters and all other initial conditions are held fixed, computation is done in double precision, and deterministic training is used. Our example further shows that minibatched training in double precision can remedy this longitudinal repeatability problem in some instances. These results inform the R&D and test and evaluation procedures that should be used for clinical-grade classifiers, and motivate further study of round-off error in the context of neural network reproducibility.

## I. Introduction

AI classifiers have demonstrated incredible usefulness for clinical diagnostics. They can distinguish between tumors that are genetically distinct but morphologically identical [1] or to stratify patient risk for numerous conditions [2]. Needless to say, a clinical classifier must be safe and accurate, and these qualities are proven by the satisfaction of numerous criteria pertaining to precision, sensitivity, specificity and repeatability. Furthermore, its implementation must also satisfy criteria pertaining to fault tolerance [3]. This paper will investigate repeatability, specifically focusing on repeatability over time—a concept we term *longitudinal consistency* and we define in Section II-A.

We aim to understand the limits of longitudinal consistency for AI classifiers, and we concentrate on neural networks, since they are powerful models with wide adoption, and we expect them to find wider use. Neural network classifiers have fundamental limitations [4] on longitudinal consistency, a consequence of the well-known nonconvexity of their loss functions. This nonconvexity renders the iterative fitting of network parameters sensitive to numerical imprecision; consequently, given a fixed training routine and training data, it may produce networks with different parameters depending on the hardware used for training. Our example demonstrates this behavior on a simple neural network that conforms to general best-principles of design [5]—including deterministic training. Our findings provide evidence that round-off error is the underlying cause.

It is important to remember that this repeatability issue has been witnessed before [4], but that earlier work considered the nondeterminism in how neural network training routines exploit GPU hardware. Our example differs in key ways: 1. our model is much smaller and simpler than the models trained in [4], 2. our training uses the software settings[1] that are supposed to disable nondeterminism in training, 3. we provide an analysis that identifies the source of divergence, and 4. from our analysis, we derive a simple method that can obtain repeatability on different hardware and demonstrate its application and limitations.

In terms of clinical relevance, our example shows that choosing a disaster recovery policy consisting of a simple from-scratch rebuild may lead to an unexpected problem: the rebuilt classifier is not the same as the one that was lost. This is significant simply by necessitating a complete model revalidation for the rebuilt classifier.

An important consideration of our analysis is that the repeatability problem we investigate here is specific to *ab initio* training of multilayer perceptrons with nontrivial depth. How these results generalize to other architectures is not guaranteed, though we note that our analysis only assumes a non-convex loss function and is applicable whenever that holds true. Likewise, those machine leaning methods which enjoy convex loss functions, such as logistic regression, support vector machines, or even neural networks constructed by iterated fitting of just one layer while others are "frozen," can always obtain repeatability up to a multi-

---

[1]`torch.use_deterministic_training_algorithms`

ple of machine tolerance due to the absence of saddle points.

We will begin by providing some preliminary definitions of classifiers, their expected clinical workflows, and their lifecycles in Section II-A. In Section II-B, we introduce longitudinal consistency and consider some cases where it matters. We then review neural network training in Section III-A and present our example—the main result—in Section IV.

## II. CLASSIFIERS AND CLINICAL WORKFLOWS

### A. Classifiers in clinical use

We present our formal definition of a classifier:

*Definition 1 (Classifier):* A **classifier** that predicts classes $\mathbf{C} = (c_1, c_2, \ldots, c_N)$ from measurements in a domain $\Omega$ is a mapping $f : \Omega \mapsto S$, with $S = \{x \in [0,1]^N \wedge \|x\|_1 = 1\}$.

A classifier produces *calls* on samples $x \in \Omega$.

*Definition 2 (Classifier calls and scores):* Let $x \in \Omega$ and $f(x) = y$. Then the **call** of $f$ on $x$ is $c_i$ with **score** $y_i$, where $i = \arg\max_j y_j$.

In clinical use, the calls and scores are compared against a critical value $\tau$ to evaluate the conclusiveness of the call. Calls with scores $y_i \geq \tau$ are deemed sufficiently conclusive to merit clinical reporting as class $c_i$, while calls with score $y_i < \tau$ are deemed inconclusive. The critical value $\tau$ remains fixed and is chosen to provide the necessary accuracy and recall prior to clinical deployment.

### B. The classifier lifecycle and rebuilds

Once integrated into clinical operations, AI classifiers may have a long deployment lifecycle. Through this lifecycle, the classifier may undergo retraining to improve accuracy, extend the codomain $\mathbf{C}$ of the classifier, or simply to upgrade software dependencies. Each retraining of the classifier results in a new *build*, even if the version of the classifier is unchanged. An example of a retraining that produces a new build but not a new version would be one that simply updates software dependencies. We enumerate these builds $f^{(1)}, f^{(2)}, \ldots, f^{(n)}$.

The criterion with which we concern ourselves is *longitudinal consistency*—the repeatability of the classifier over time. More formally: A series of classifiers $(f^{(i)})_{i=1}^{n}$ is **longitudinally consistent** on test set $S \subset \Omega$ at confidence $0 < p \leq 1$ if at least $p$ proportion of the time, all builds $f^{(i)}$ give the same result whenever they all produce clinically-reportable results. Note that the appropriate value for $p$ is problem-dependent, and should be determined by a risk analysis that considers the consequences of inconsistency in classifier results.

Satisfaction of longitudinal consistency with large $p$ is particularly important when the classifier is used for:

1) Stratification of patients in a long-lasting clinical trial, during which both rebuilds of the classifier and retrospective analyses occur.
2) Serial monitoring of patients over time to detect response to treatment, during which rebuilds of the classifier may occur.
3) Implementation of a disaster recovery policy due to corruption or loss of the classifier's storage files.

Longitudinal consistency is important for Item 1 so that retrospective analysis results do not change with new builds $f^{(i)}$; for a long-lasting trial, discrepancies between builds may require exclusion of a patient from the trial's results—a serious concern for trials on rare conditions. Likewise, for Item 2, longitudinal consistency ensures that the serial monitoring is coherent; discrepancy between builds presents the conundrum to the treating physician: "which call is correct?" Item 3 requires longitudinal consistency to mitigate the risk that the rebuilt classifier $f^{(n)}$ must be completely re-validated because it is too discrepant with the already-validated prior build $f^{(n-1)}$—this is the unexpected problem we mentioned earlier.

The clinical consequences of longitudinal consistency failures can be significant; for example, Group 3 and Group 4 Medulloblastomas present a classification problem of similar difficulty to the Gliobastoma problem presented in Section IV. Group 3 tumors respond to carboplatin treatment, but Group 4 tumors do not [6]. Carboplatin is ototoxic, so its use should be restricted to Group 3 tumors. Without reliable Group 3/4 classifications, this distinction could be useless.

## III. TRAINING NEURAL NETWORKS AND FINITE-PRECISION ARITHMETIC

### A. Neural network fundamentals: architecture

As we have mentioned previously, many clinical classifier developers will select neural networks as their first choice for a new application, and with good reason [7]. While there is a wide variety of neural network architectures, overall they can be described as an iterated composition of functions as:

*Definition 3 (Neural Network):* A neural network on $\Omega$ is a series of functions $(g_i(x; P_i))_{i=1}^{m}$ with parameters $P_i$ composed as

$$f(x) = g_m\left(g_{m-1}\left(\cdots g_1(x; P_1) \cdots ; P_{m-1}\right); P_m\right)$$

The functions $g_i(x; P_i)$ are usually called the *layers* of the neural network, and could include linear transformations, nonlinear activation functions, normalization operations, quantization or pooling operations, among others. The parameters $P_i$ are discovered by iterative minimization of a specific loss function $\mathfrak{f}(f(x), b)$, where $b \in \mathbf{C}$ is the ground truth for sample $x$. The iterative minimization requires an ansatz—often simply random values [8]. These parameters $P_i$ define the network and are fixed at training time. Minimization

continues until a stopping criterion is satisfied. While many suggestions exist for the stopping criterion, the magnitude of the gradient[2] $\nabla_{(P_1,\ldots,P_m)}\mathfrak{f}(f(x),b)$ is not one of them [5].

Gradient descent is a popular choice for training neural networks and many variations exist. Updated guesses $\hat{P}_i$ for the parameters $P_i$ follow the update rule

$$\left(\hat{P}_1,\ldots,\hat{P}_m\right) \leftarrow (P_1,\ldots,P_m)$$
$$+ \alpha\tfrac{1}{k}\sum_{j=i}^{i+k} \nabla_{(P_1,\ldots,P_m)}\mathfrak{f}(f(x_j),b_j) \quad (1)$$

for $x_j \in S_{\text{train}}$ and $b_j \in \mathbf{C}$, and partial differentiation is performed with respect to the parameters $P_i$. The weight $\alpha$ is algorithm-dependent. Note that $i$ in (1) is most often chosen to loop over the training set $S_{\text{train}}$ with stride $k$. Here, $k$ is the size of the "minibatch" and $k = 1$ for stochastic gradient descent. The step size $\alpha$ is algorithm dependent. We also omit any scaling of the gradients $\nabla_{(P_1,\ldots,P_m)}\mathfrak{f}(f(x_j),b_j)$ to balance class sizes.

The loss function is typically equivalent to negative log likelihood; due to the compositional structure of $f$, it is usually non-convex. Naturally, this non-convexity persists even if a convex regularization penalty is added, such as an L2 penalty—often called "weight decay." Non-convexity implies that the loss function has local minima which will prevent iterative minimization methods from ever finding the global minimum.

While the non-convexity of the loss function is well known, it is widely understood that the presence of local minima is not a problem due to the "no bad local minima" property exhibited by most neural networks [9], [10]. In addition to local minima, non-convex functions also can have saddle points, which are $x \in \Omega$ that have

$$\|\nabla_{(P_1,\ldots,P_m)}\mathfrak{f}\| = 0 \quad (2)$$
$$\lambda_{\min}\left(\mathbf{H}_{(P_1,\ldots,P_m)}\mathfrak{f}\right) < 0 < \lambda_{\max}\left(\mathbf{H}_{(P_1,\ldots,P_m)}\mathfrak{f}\right) \quad (3)$$

with gradient $\nabla_{(P_1,\ldots,P_m)}\mathfrak{f}$ and Hessian $\mathbf{H}_{(P_1,\ldots,P_m)}\mathfrak{f}$ with minimum and maximum eigenvalues $\lambda_{\min}(\mathbf{H}_{(P_1,\ldots,P_m)}\mathfrak{f})$ and $\lambda_{\max}(\mathbf{H}_{(P_1,\ldots,P_m)}\mathfrak{f})$, respectively. What distinguishes them from local minima is (3). In plain language, saddle points are points where the gradient vanishes and are also flanked by neighborhoods where the loss function curves *both* upward and downward; they are the multidimensional analogue to mountain passes. In infinite-precision arithmetic, saddle points would also trap iterative solvers, but with finite precision, it is almost surely the case that round-off errors make (2) impossible.

### B. Neural networks and finite-precision

Finite precision arithmetic issues are not a usual consideration for neural network training. This is because neural network training does not aim to drive

---

[2]after this we will omit the $(f(x),b)$ in writing the gradient and Hessian when it is not ambiguous

the gradient $\nabla_{(P_1,\ldots,P_m)}\mathfrak{f}$ to small values for neural networks. In fact, double precision arithmetic is often not used, because it is not well-supported by all graphics cards and does not add and a benefit [11]. But if one is trying to replicate parts of a neural network for longitudinal consistency, floating point arithmetic can make a difference.

The main way floating point precision impacts neural network training is in the computation of the gradient. Gradient values $\nabla_{(P_1,\ldots,P_m)}\mathfrak{f}$ will contain error, so saddle points become saddle neighborhoods; that is, (2) becomes

$$\|\nabla_{(P_1,\ldots,P_m)}\mathfrak{f}\| =_\epsilon 0 \quad (4)$$

where $=_\epsilon$ is floating point equality up to machine precision $\epsilon$, which in turn depends on the precision of the arithmetic that is used.

To see when and how the gradient numerically vanishes—i.e. (4) is satisfied, we use a linearized approximation of the gradient at an expansion point $(P_1,\ldots,P_m)$ to find $\nabla_{(P_1,\ldots,P_m)+\varepsilon}$ for an offset $\varepsilon$

$$\nabla_{(P_1,\ldots,P_m)+\varepsilon}\mathfrak{f} \approx \nabla_{(P_1,\ldots,P_m)}\mathfrak{f} + \mathbf{H}_{(P_1,\ldots,P_m)}\mathfrak{f}\varepsilon$$

which, combined with (4) gives

$$\left(\mathbf{H}_{(P_1,\ldots,P_m)}\mathfrak{f}\right)^{-1}\nabla_{(P_1,\ldots,P_m)}\mathfrak{f} =_\epsilon \varepsilon; \quad (5)$$

that is, the error $\varepsilon$ needed to annihilate gradient and push us into a saddle neighborhood is exactly $\left(\mathbf{H}_{(P_1,\ldots,P_m)}\mathfrak{f}\right)^{-1}\nabla_{(P_1,\ldots,P_m)}\mathfrak{f}$.

This observation gives us the "perfect storm:"

• saddle points expand from singularities to neighborhoods,

• the almost-universal presence of error means the gradient never vanishes—it is only non-error digits that vanish.

If gradient descent ever encounters a saddle neighborhood, then the parameter update will be dominated by error. The error can and does vary from machine to machine, so differences in the floating point error of different computers can cause different updates in (1). Differing updates in a saddle neighborhood can cause rapid divergence; a training routine that is deterministic in every other way can produce different neural networks when different hardware is used.

*Remark 1:* Depending on the stability of the gradient computation at $(x,b)$, $\epsilon$ may be bigger than the machine's floating point tolerances; for example, catastrophic cancellation [12] can cause errors of magnitude much larger than machine epsilon.

*Remark 2:* Saddle neighborhoods are impossible to rule out, and may even be likely if there are training samples $x_i$ that the classifier can predict nearly perfectly early on in the minimization process.

*Remark 3:* (5) shows that errors $\varepsilon$ with small norm will happen when the Hessian has large singular value

triples $(u_j, v_j, \sigma_j)$ for $j = 1, 2, \ldots, k$ and $\nabla_{(P_1,\ldots,P_m)}\mathfrak{f}$ is well-approximated in span$\{u_1, \ldots, u_k\}$.

*Remark 4:* Based on Remarks 2 and 3, using a mini-batch size $k > 1$ may help mitigate the risk of encountering saddle neighborhoods. This is because all of $\nabla_{(P_1,\ldots,P_m)}\mathfrak{f}(f(x_i), b_i)$ have to align in span$\{u_1, \ldots, u_j\}$ for $k$ of the $x_i$, but also because the summation of the Hessian will be the arithmetic mean of $k$ Hessians from (5). This averaging will decrease the magnitude of the singular values due to Weyl's inequality for singular values [13]—i.e. $\sigma_1(A + B) \leq \sigma_1(A) + \sigma_1(B)$.

## IV. THE MAIN RESULT

We are now ready to present our main result. First, we selected a problem with clinical application, and designed a neural network to classify it. We followed best-practices design principles for network architecture, hyperparameter selection, and training, so that the models are a reasonable representation of the classifier that would be deployed clinically. The only deviation from this principle is that we trained in double-precision in addition to single-precision in an effort to minimize the magnitude of round-off error. This model failed longitudinal consistency whenever the model was rebuilt on different hardware in both precision types.

The non-repeatability does not happen when the classifier is repeatedly trained on the same hardware. To confirm the suspicion that floating-point error is to blame, we investigate the local curvature of the loss function throughout iterates of the training. We also investigate a fix motivated by Remark 4: setting $k > 1$, which remedies the repeatability problem for double-precision arithmetic. For single precision, repeatability is still problematic; we treat this issue separately.

First we define the classification problem we will solve, and then the network architecture and training routine. For $k = 1$, the results show:

- networks trained on the same hardware are bitwise identical, confirming that the training routine is deterministic in software,
- the parameters $P_i$ of the networks trained on different hardware diverge over training iterations,
- the divergence is rapid and sudden—apparently not due to gradual accumulation of error in the parameters $P_i$, but due to a catastrophic update step (1) during training,
- the divergence happens while the model is still generalizing on the training data—overfitting was not yet apparent,
- the path traversed by gradient descent gets increasingly close to saddle neighborhoods as training proceeds.
- the difference between model trained on different hardware is big enough to have clinical impact—they

are not longitudinally-consistent even for non-trivial values of $p$.

The observations imply that it is round-off error that is responsible for the divergence in training and the failure to get the same neural network on different hardware. The last observation is especially problematic: it creates longitudinal consistency problems.

### A. Hardware and libraries used

The data, hardware and software is as follows: a Dell dimension 7960 workstation with an NVIDIA RTX 4500 and an Intel Xeon processor, and a HP Apollo server with an NVIDIA A100 and a AMD Ryzen processor, the classifier was implemented in python 3.11 [14], with Pytorch [15]. Anaconda [16] is used to manage versions of all software dependencies, `git` was used for version control of python code and YAML specifications of the conda environment. Models trained on the Dell workstation CPU and GPU will be labeled as "Xeon" and "RTX,"respectively. Models trained on the HP server will be labeled as "Ryzen" and "A100,"respectively.

### B. Problem description

Our problem is to classify a subset of the DNA methylation classification problem posed in [1]. We train only to recognize the methylation classes in the "GBM" family: "GBM, G34," "GBM, MID," "GBM, MES," "GBM, MYCN," and "GBM, RTK I-III." These represent one of the more difficult groups of CNS tumors to classify, since they share many genetic similarities but are still distinct. The classifier does not include any control tissue.

We deploy this classifier in a mock clinical application to distinguish between the RTK I and all others, so all non-RTK I classes are lumped into a "non-RTK I" class for a straightforward computation of the clinical cutoff $\tau$ from Section II-A. This lumping was only for computing $\tau$, and we still report calls for all 7 classes. The supposed purpose of this classification is to identify the RTK I tumors for a novel therapy that is a deescalation from the standard of care that shows promise for RTK I. In the mock clinical application, we require longitudinal consistency for some small $p$ due to the inherent risk in deescalating from standard of care, though we leave this variable unbound. We just present it for the sake of analyzing the similarity of the different neural networks we have trained.

[1] provides two data sets, a training set and an independent test set; this allows us to simulate a clinical deployment with longitudinal consistency evaluations on an independent clinical set. The set GSE90496 was used for training and GSE109379 was used for longitudinal consistency evaluation. Only the "GBM" classes samples from GSE90496 were used for training, and only "GBM, RTK I" samples from GSE109379 were used for the longitudinal consistency analysis.

## C. Neural network architecture and training routine

To classify our methylation problem, we built a simplified version of the neural network described in [17]; we omitted the sparse first layer, dropout layers, and the bottleneck layers. That gave us a network with 4 fully-connected layers with dimensions 51,000–128–32–7, with interleaved layer normalization and softshrink activation functions. We also used the same feature selection as in [17], but we used M-values instead of beta values for better potential performance [18]. We trained the model with Adamax [19], with a variable minibatch size (c.f. Remark 4 & [5]), and learning rate determined by a brute-force hyperparameter optimization.

## D. Hyperparameter optimization

Adamax requires weight decay and learning rate parameters. To find these values, we used 64-bit models and performed a brute-force grid search of learning rate and $L2$ penalty weight spaces on a $8 \times 8$ logarithmically-sampled grid. The grid endpoints were $\exp(-6)$ and $\exp(-2)$ for both the $L2$ penalty and learning rate. We split the GSE90496 training data 75%-25% using stratified downsampling, keeping the 25% partition for evaluating performance. For each gridpoint, we 1. trained a neural network on 64 iterations of out training algorithm, 2. during training, evaluated the in-training model at the end of each epoch and recorded the error rate at full recall, 3. recorded the minimum error obtained over all epochs of training. Note that while we trained each network for 64 epochs, we evaluate the "success" of each gridpoint based on the best epoch's error rate on the evaluation data, which is not necessarily the 64th epoch. The learning rate and $L2$ penalty that produced the lowest error rate is used for training the neural networks in the reproducibility test that followed.

## E. Repeated builds on different hardware

So that our test of repeatability is valid, it is crucial to hold all sources of nondeterminism fixed and ensure that all dependency versions are identical—this way the only difference between the builds is the hardware. To do this, we used fresh conda environments built to the same specification and ensured that all random number sequences were seeded with the same value. To verify our build process is repeatable on the same hardware, we repeat each build process twice. We used double-precision floating point types in an attempt to mitigate the influence of round-off error, and repeated single precision only on "A100" and "Ryzen." To test repeated builds on different hardware and precision, we:

1) cloned the same snapshot of the `git` repository on each machine used.
2) built a new conda environment using `conda env` on the YAML environment specification stored in the repository
3) for each machine in Section IV-A, repeated the following steps twice, using `cpu` as the `pytorch` device:
   a) set the seed of the random number generators for both pytorch and numpy and initialized a neural network with default pytorch settings.
   b) selected the same cross-validation split used in hyperparameter optimization,
   c) trained the neural network using the optimal hyperparameter values from Section IV-D, $k = 1$, saving copies of the parameters and gradients at numerous points in the iterations.
4) repeated step 3 using `cuda` as the `pytorch` device.
5) for each model, use the holdout data to perform a ROC analysis to determine the clinical cutoff $\tau$ for the mock analysis.

As we had mentioned in Remark 4, we also want to show that a minibatch size greater than 1 repairs our repeatability problem. For that, we simply repeated the above steps with $k \geq 2$.

To judge the impact of the differences between the models, we measure both the dissimilarity between the parameters that define the models and longitudinal consistency on GSE109379. We measure parameter dissimilarity with vector sine angles: $\sqrt{1 - \cos \angle(x, y)^2}$. We measured longitudinal consistency by applying the mock clinical analysis protocol described in Section IV-B and counting the number of cases with discrepant calls.

## F. Evaluation of longitudinal consistency with $k = 1$

In our repeatability test—the minibatch size $k = 1$ and 64-bit precision, we saw that the same-hardware replicates were bitwise identical. However, all 4 different hardware devices produced a unique neural network. The clinical impact is nontrivial: in our mock clinical deployment, the "Xeon" and "RTX" models were discrepant on 4% of all clinically-reportable calls on GSE109379, using only the samples that are in the domain $C$ for this problem—the "GBM" classes. The implication of this is that the attempted model retrain failed, and that the newly-trained model must be subject to an entirely new validation; moreover, there is risk that patients may have been incorrectly subjected to deescalated therapy.

The mock clinical analysis is meant to simulate a classifier rebuild that does not change the version number; for example, this could be a simple library update or implementation of a naive disaster recovery protocol. Note that this analysis is meant to evaluate model non-repeatability, so the model replication training is simply a recapitulation of the original training.

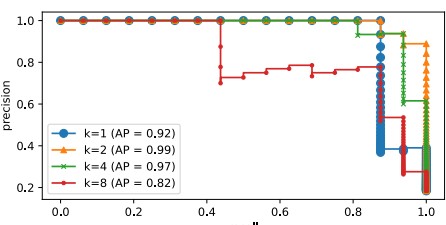

Fig. 1. Precision and recall on holdout set for $k = 1, \ldots, 8$ on 64-bit models. The clinical cutoff chosen to maximize recall subject to perfect precision on the holdout set is $\tau = 0.3$ for $k = 1$. Note that increasing the clinical cutoff would have no improvement to precision on the holdout set but would result in lost recall. All models were trained for 64 epochs, except for $k = 8$ which exhibited overfitting and was stopped early at epoch 59.

This means that fair comparisons of precision and accuracy on the holdout set is not entirely possible— if the replicate has indeed diverged from the original model, then it was not necessarily trained with the correct early stopping and its clinical threshold may be substantially different. However, the attempted replicate model represents the best attempt at model replication. We also note that hold-out error did not indicate overfitting for the recapitulated model.

We first simulate the initial build of the classifier and set the clinical reporting cutoff $\tau = 0.3$, based on an ROC analysis on the classifier output on the 25% holdout partition (defined in Section IV-D and used for both hyperparameter optimization and training). The ROC analysis simply picked the score that maximized recall subject to obtaining perfect precision. For the ROC analysis, the classification problem was coerced into a binary classification problem by summing all non "GBM, RTK I" classes into a synthetic class "not GBM, RTK I." The clinical mock analysis used all classes in **C**. We then consider that the classifier is rebuilt for some reason and must pass a retrospective proficiency test. We assume that the original model was "RTX" and rebuilt model was "Xeon" for convenience. We then counted all cases that failed to satisfy longitudinal consistency.

The model obtained full recall on the 339 "GBM" samples in GSE109379, and over 4% of these cases produced different calls that were clinically-reportable. One might be tempted to simply adjust the score cutoff $\tau$ upward to prevent failures in longitudinal consistency, but there is no way to know that $\tau$ *a priori.*

The relationship between precision and recall on the holdout set is shown in Figure 1 for $k = 1, 2, 4, 8$ to illustrate the influence of minibatch size on performance for 64-bit models; The precision and recall were calculated on the "RTX" model for $k = 1$ and with A100 for $k > 1$. We note that the cutoff needed to eliminate discrepancies on GSE109379 would result in moving from a recall of 0.875 to 0.8125 in Figure 1, if one knew that shift in advance.

We have not addressed precision on GSE109379; a definitive precision analysis is not possible because GSE109379 does not have actual ground truth labels. Its intended use is as a prospective clinical set; in [1], it was used for a mock clinical demonstration of a CNS tumor classifier in which calls are compared against orthogonal clinical findings. Moreover, as we have mentioned, the attempted replicate classifier may not be optimally trained since its stopping parameter and $\tau$ are borrowed from the training of the original model.

The failure of longitudinal consistency motivated the question: "how much do the models differ, and when did they become different?"

### G. Evaluation of divergence when $k = 1$ and repeatability when $k = 2, 4, 8$ for 32 and 64-bit precision

The longitudinal analysis only considered the "Xenon" and "RTX" models, we also determined if the "Ryzen," "Xenon," "RTX," and "A100" models diverged, including both 32- and 64-bit precision. Overall, $k \geq 2$ produced reproducible models in double precision, but 32-bit precision models diverged eventually, but $k = 8$ mitigated divergence best.

We analyzed model divergence by comparing the parameters $(P_1, \ldots, P_m)$ that we saved over iterates of our training algorithm for dissimilarity. We also measured the distance between the model state and saddle neighborhoods at each minibatch—this is meant to simulate if round-off error might have caused divergence. To measure dissimilarity between states, we used the saved model parameters $(P_1, \ldots, P_m)$ and computed vector sine.

To measure distance to saddle neighborhoods, we computed $\|U^\mathsf{T} \nabla_{(P_1, \ldots, P_m)} \mathfrak{f}\|$, for left singular vectors $U = [u_1 \cdots u_{64}]$ of the Hessian $\mathbf{H}\mathfrak{f}$. This approximates how much of the gradient can be annihilated by noise. The singular vectors were approximated with a block power iteration scheme; we also bounded the smallest Hessian eigenvalue from above to verify the Hessian was indefinite—this was always true.

The 64-bit models trained with $k = 1$ on different hardware were not identical, but maintained low dissimilarity in early iterations of training. They all experience rapid divergence around epoch 30. Figure 2 shows the state dissimilarity sines between "Ryzen" and "A100" when $k = 1$ and $k = 2$ for 64-bit, and $k = 1$ and $k = 8$ for 32-bit. These results show that the 32-bit models diverge at the onset of training, but $k = 8$ guards against further divergence; there is also evidence of slow accumulation of error. We refrain from plotting other 32-bit models, since they all exhibit the same divergence at epoch 1.

The 64-bit models trained with $k \geq 2$ on different hardware were not always bitwise identical, but exhibited tiny vector sines at the end of training, indicating closeness up to machine precision. Every 32-bit model diverged eventually, which we explain

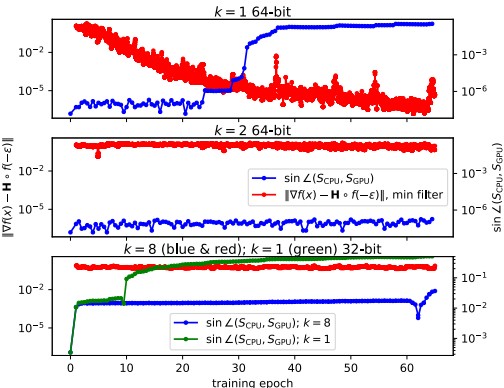

Fig. 2. Vector sine dissimilarity measurements between "Ryzen" and "A100" models (blue) and gradient annihilation residuals (red) for 64-bit models; 32-bit $k = 1$ (sine only, green) and $k = 8$ are also presented. Values are computed immediately after model state update, so epoch 0 indicates initial vector sine. Model states were only saved for some Adamax iterates, but gradient annihilation residuals were computed at every iterate. Gradient annihilation residuals are passed through a rolling neighborhood minimum filter with neighborhood size 23 to reduce high-frequency content. Increasing values of vector sine indicate increasing divergence. The divergent models are identical up to machine tolerances until rapid divergence is encountered. In 64-bit, the apparent sufficient residual value for divergence is 1e-4. Both 32-bit models exhibit divergence at epoch 1, but $k = 8$ mitigates further divergence. The square root used in computing vector sine implies that machine noise is $\leq \sqrt{\epsilon_{\text{machine}}}$.

by considering the apparent residual magnitude sufficient to encounter a saddle neighborhood in 64-bit precision—approximately 1e-4 from Figure 1. Adjusting upward by 1e3 for the lower machine epsilon of 32-bit means most residuals are sufficient to push the model state perilously close to a saddle neighborhood. This explains the immediate divergence witnessed for the 32-bit model in Figure 1.

Note that our divergence analysis does not consider the classification performance of the models, but the precision/recall curves in Figure 1 reproduce the well-known relationship between minibatch size and performance: increasing minibatch size may sacrifice AUC.

## V. Conclusion

It is already known that neural network training can be sensitive to round-off error—clearly, these reproducibility problems are to be avoided for models that must be repeatable. Round-off error is an unfortunate reality in all numerical computation, even in well-accepted CPU implementations of BLAS [20]. Since round-off error cannot be avoided, these repeatability problems will show up unless the hardware used for training can be matched exactly. Over long timescales, exact matching of hardware becomes harder to the point of infeasibility. Clinical classifiers may have long lifecycles, so we should assume that matching hardware over a classifier's lifecycle is not practical.

What we have shown here is that the repeatability problem can be addressed by using a minibatch with 64-bit precision, and for our example, a minibatch size

of 2 was sufficient. Our approach was motivated by a brief analysis of a linearized model of the curvature of the loss function $\mathfrak{f}$. When the Hessian $\mathbf{H}\mathfrak{f}$ becomes numerically low-rank and the gradient $\nabla \mathfrak{f}$ is well-approximated in the numerical column space of $\mathbf{H}\mathfrak{f}$, then we can encounter saddle neighborhoods due to error. Our analysis suggested that minibatches address the reproducibility problem by an averaging process over the gradient and the Hessian; when the samples in the minibatch are not too colinear, then the averaged gradient may be less likely to fall completely in the numerical column space of $\mathbf{H}\mathfrak{f}$, and the operator norm of $\mathbf{H}\mathfrak{f}$ is also likely reduced, but for lower precision formats, this reduction may not be enough to completely offset larger $\epsilon_{\text{machine}}$ and loss of guard digits.

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
