# OpenReview forum: "Minibatches can make neural network training repeatable for clinical applications"
_IEEE.org/EMBS/BHI/2025/Conference — BHI 2025_

### Official Review · Reviewer_zXxM · 2025-06-28
**Minibatches can make neural network training repeatable for clinical applications**

**Confidence:** 4
**Clarity Of Writing:** excellent
**Clinical Significance:** great
**Methodological Novelty:** good
**Overall Rating:** 7

**Experiments And Results:**

great

**Questions For The Authors:**

Generalizability:
Question: Have you tested this repeatability issue and the minibatch solution on other neural architectures or clinical datasets beyond DNA methylation classification?
Impact: Demonstrating broader applicability could significantly strengthen the generality and clinical value of the findings.

Sensitivity to Minibatch Size:
Question: Did you explore whether larger minibatch sizes (e.g., k = 4, 8, 16) have additional benefits or drawbacks? Could there be a trade-off in performance or convergence behavior with increasing batch size?
Impact: Insight into optimal batch size would improve the practical utility of your recommendation and affect methodological assessment.

Impact on Low-Precision Training and Deployment:
Question: Have you considered how these reproducibility issues behave under low-precision training (e.g., float16) or quantized inference, common in deployment settings?
Impact: Extending your results to realistic deployment conditions would significantly increase relevance and real-world impact.

Formal Definition of Acceptable Divergence:
Question: How should practitioners determine what level of divergence is acceptable in clinical applications? Could you propose a general guideline for longitudinal consistency thresholds?
Impact: Addressing this would help operationalize your findings in real clinical workflows and increase the clarity and applicability of your contribution.

**Strengths:**

The paper addresses longitudinal consistency, a critical but underexplored requirement for deploying AI models in clinical settings, especially for longitudinal monitoring, clinical trials, and disaster recovery.

Demonstrates that even under deterministic settings and double-precision arithmetic, training results can diverge across hardware due to round-off error, this is rarely quantified so clearly.

Proposes a simple and effective fix, minibatch size >1, backed by both theoretical analysis (saddle neighborhoods, Hessian properties) and empirical evidence.

Repeats training on multiple hardware setups with controlled environments and shows reproducibility failures and recovery through systematic experiments.

Provides an insightful and elegant linearized analysis of how round-off error interacts with saddle points in training, explaining why minibatches mitigate divergence.

Clear documentation of environment control, hardware/software setup, and training configurations supports full reproducibility of findings.

**Summary Of The Paper:**

It's a well-written and technically rigorous study that identifies and analyzes the critical issue of longitudinal inconsistency in neural network training due to round-off errors—even under deterministic settings and double precision. The paper shows that using minibatches (k > 1) can effectively mitigate this issue, making it highly relevant for clinical deployment scenarios where repeatability is essential.

**Weaknesses:**

The findings are demonstrated on a specific DNA methylation classification task. It's unclear how generalizable the results are across diverse architectures, data modalities, and clinical problems.

While the paper shows that k = 2 improves repeatability, it does not explore how minibatch size affects training speed, convergence, or final model performance more broadly.

The paper shows a 4% discrepancy rate as clinically meaningful, but doesn’t propose a generalizable threshold for acceptable divergence in other clinical use cases.

Many real-world clinical models deploy in 16-bit or 8-bit precision for speed and memory efficiency. It’s unclear whether the conclusions hold in those regimes.

Although the minibatch fix maintains clinical accuracy in this example, the paper doesn’t formally evaluate whether there's a broader performance trade-off (e.g., overfitting, generalization loss).

---

### Official Review · Reviewer_3sPK · 2025-07-04
**Review on "Minibatches can make neural network training repeatable for clinical applications"**

**Confidence:** 5
**Clarity Of Writing:** good
**Clinical Significance:** great
**Methodological Novelty:** great
**Overall Rating:** 7

**Experiments And Results:**

great

**Questions For The Authors:**

### **Questions for the Authors**

1. **Generalizability of the Minibatch Fix**:
   *Question*: Have you tested the proposed minibatch remedy (e.g., $k = 2$) on other neural network architectures (e.g., convolutional or transformer-based models) or tasks beyond the CNS tumor classification problem?
   *Why it matters*: If this fix is broadly effective, it significantly elevates the impact and relevance of the work across machine learning. A demonstration of generality would strengthen the claim that minibatching is a broadly applicable strategy for ensuring training repeatability.
   *Score impact*: A positive answer here would raise the novelty and applicability score.

2. **Minibatch Size Sensitivity**:
   *Question*: How sensitive is the repeatability outcome to the specific choice of minibatch size? Is there a minimum $k$ beyond which reproducibility stabilizes?
   *Why it matters*: Knowing whether there’s a threshold effect or a gradual improvement with increasing $k$ can guide practitioners in choosing the smallest effective minibatch size, especially under resource constraints.
   *Score impact*: If the authors provide this data or analysis, it would improve the rigor and completeness evaluation.

3. **Behavior of Gradient Residuals Near Divergence**:
   *Question*: In Figure 2, the residuals leading to divergence increase, but have you characterized any consistent pattern or condition just before the divergence point? Could this be used predictively?
   *Why it matters*: Understanding the early signs of divergence might allow for monitoring and intervention before a reproducibility failure occurs, which would be extremely useful in clinical pipelines.
   *Score impact*: A strong predictive signal would enhance the practical utility of the work.

4. **Model Performance Across Builds**:
   *Question*: Beyond call discrepancies, did you observe any differences in traditional performance metrics (AUC, accuracy) across builds trained on different hardware?
   *Why it matters*: While 4% clinical call divergence is nontrivial, it would be useful to know if this translates into statistically significant performance differences.
   *Score impact*: If differences are systematic, it would reinforce the severity of the reproducibility issue and justify the need for the proposed remedy.

5. **Relevance to Training with Mixed Precision**:
   *Question*: Since clinical AI workflows may favor efficiency, have you considered the implications of your findings under mixed-precision training (e.g., FP16/FP32), which is common in modern deployments?
   *Why it matters*: If the reproducibility problem worsens under mixed precision and minibatching still helps, it would further validate your proposal in a practical setting.
   *Score impact*: An answer here could extend the paper’s relevance to cutting-edge deployment practices.

**Strengths:**

1. **High Clinical Relevance**: The paper addresses a practical and critical problem—ensuring model repeatability in clinical settings—where inconsistencies can directly affect patient care and clinical trial outcomes.

2. **Novel and Actionable Insight**: The discovery that using minibatches (even as small as size 2) can mitigate hardware-induced non-repeatability is both novel and immediately applicable for practitioners deploying neural networks in sensitive domains.

3. **Rigorous Experimental Design**: The authors meticulously control for all known sources of nondeterminism—random seeds, software versions, data splits, precision settings—ensuring that observed differences are genuinely due to hardware-specific round-off errors.

4. **Theoretical Justification**: The analysis linking saddle neighborhoods and gradient annihilation to divergence provides a sound theoretical explanation for the observed phenomena and supports the minibatching fix.

5. **Empirical Depth**: The paper supports its claims with a comprehensive suite of empirical evaluations, including ROC-based cutoff analysis, vector sine similarity of parameters, and gradient annihilation metrics.

6. **Use of Real Clinical Data**: Employing a challenging, clinically significant classification task (CNS tumors) and real-world datasets enhances the paper’s credibility and applicability.

7. **Clarity in Definitions and Motivation**: The introduction of “longitudinal consistency” as a formalized concept with defined clinical implications is clear and well-motivated.

8. **Reproducibility Emphasis**: The work adheres to best practices in experimental reproducibility, including version control and controlled environments, aligning with open science and clinical transparency values.

**Summary Of The Paper:**

The paper investigates a critical but underexamined issue in the clinical deployment of AI classifiers: **longitudinal consistency**, which refers to the ability of repeated builds of a classifier to produce consistent outputs over time—even when retrained on the same data and with fixed software parameters.

The authors demonstrate that even under deterministic settings in PyTorch, double-precision arithmetic, and identical software and hyperparameters, training a neural network on different hardware (CPUs and GPUs from different vendors) can lead to **non-reproducible model parameters and inconsistent outputs**, due to floating-point round-off errors.

They identify that this divergence occurs when the training trajectory encounters **saddle neighborhoods** in the loss landscape. At these points, small hardware-specific numerical differences in gradient computations lead to diverging optimization paths. This divergence can affect clinically-reportable decisions, potentially undermining the integrity of clinical workflows that depend on classifier stability.

To address this, the authors propose using **minibatches with size greater than one (e.g., $k=2$)**. Through empirical tests and theoretical justification, they show that minibatching reduces the chance of catastrophic divergence by averaging out error-prone gradients and Hessians, thus improving **repeatability across different hardware platforms**.

The paper provides experiments using a methylation-based CNS tumor classification task, comparing builds across four hardware configurations. Results show that while training with $k=1$ leads to divergence and up to 4% disagreement in clinical calls, training with $k=2$ eliminates these discrepancies without sacrificing performance.

In conclusion, the authors highlight that **minibatching can act as a simple, effective strategy to ensure repeatable neural network training for clinical AI systems**, where consistency across time and hardware is essential.

**Weaknesses:**

### **Weaknesses**

1. **Limited Scope of Generalization**:
   The experimental validation is confined to a single type of model (a feedforward neural network) on a single domain-specific task (methylation-based CNS tumor classification). It is unclear whether the observed benefits of minibatching extend to other architectures (e.g., CNNs, transformers) or domains (e.g., vision, NLP, tabular data). Broader applicability would strengthen the impact of the findings.

2. **Minibatch Size Justification**:
   While minibatching with $k=2$ is shown to resolve the reproducibility issue, the choice appears arbitrary. There is no analysis of the sensitivity of model repeatability to varying minibatch sizes. A small experiment showing how repeatability or divergence trends with increasing $k$ could provide more robust guidance.

3. **Lack of Formal Quantification of "Saddle Neighborhoods"**:
   While the authors propose a metric for saddle neighborhood proximity based on gradient projection onto Hessian singular vectors, they do not establish a formal divergence threshold or offer a predictive model based on this metric, which could enhance its practical utility.

4. **No Tradeoff Analysis**:
   While the paper asserts that minibatching doesn't degrade clinical performance, there is no systematic analysis of the trade-offs between batch size, model generalization, and training time/resource usage. This would be particularly relevant for models trained in resource-constrained environments.

5. **Precision Estimation Limitations on Test Set**:
   The test set (GSE109379) is used for evaluating longitudinal consistency, but the authors admit the ground truth labels are incomplete, preventing full precision calculation. Although the motivation is clear, the lack of precision limits the ability to fully assess the clinical impact of model discrepancies.

6. **Reproducibility vs. Robustness**:
   The study focuses on repeatability (training deterministic models that are hardware-invariant), but it does not address whether these models are robust to data distribution shifts or adversarial conditions—factors equally important for clinical reliability. This distinction could be better discussed.

---

**Suggestions for Improvement**:

* Include a broader set of model architectures or tasks to test generality.
* Perform a minibatch ablation (e.g., $k = 1, 2, 4, 8$) to explore the threshold effect.
* Quantify “saddle neighborhood proximity” with more rigorous or visual metrics.
* Add a discussion on trade-offs between reproducibility, resource usage, and generalization.
* Consider integrating simulations or surrogate labels to better estimate precision in test evaluations.

---

### Official Review · Reviewer_m7FH · 2025-07-16
**Longitudinal consistency for clinical grade AI - strong methodology**

**Confidence:** 4
**Clarity Of Writing:** great
**Clinical Significance:** great
**Methodological Novelty:** great
**Overall Rating:** 7

**Experiments And Results:**

great

**Questions For The Authors:**

1. How sensitive is the fix (minibatch size k > 1) to the value of k? Would k=4 or k=8 offer even greater stability?
2. Have you observed similar reproducibility issues in other clinical model types? You can add a discussion section on this.

**Strengths:**

1. The paper addressed an important and often overlooked requirement of the clinical-grade AI to have the ability to produce consistent results over time.
2. They offered a clear theoretical explanation of the unstable convergence. Also, how training a classifier with k=1 on different hardware yields clinically significant discrepancies. In this case, the “CPU-Xeon” and “GPU-RTX” models were discrepant on 4% of all clinically-reportable calls on GSE109379, using only the samples that are in the domain C for this problem—the “GBM, IDHwt” family.
3. The paper showed how increasing minibatch size to k=2 resolves divergence without loss of performance, which is a minimal but effective solution.
4. The software and hardware setup were very well documented. The identical hardware replication confirms that the issue was originating from the floating-point error, not randomness.
5. Finally, to prove this point, they used a clinically relevant case study.

**Summary Of The Paper:**

The paper addressed an important and often overlooked requirement of the clinical-grade AI to have the ability to produce consistent results over time. The paper demonstrates that the round-off error, even under supposedly deterministic conditions in PyTorch, leads to divergence between models trained on different hardware. They also proposed and verified that using minibatches with k > 1 may help mitigate the risk of encountering saddle neighborhoods and improve repeatability.

**Weaknesses:**

1. It’s unclear how general this solution is across other architectures.
2. All experiments are on a single network and classification task. This limits conclusions about how common this problem is across model types or domains. You can consider this in future work, or include a note in the discussion.

---

### Official Review · Reviewer_mgqZ · 2025-07-18
**A valuable and well-motivated paper on clinical AI repeatability, but its narrow scope and lack of comparison to alternative methods limit the generalizability and depth of its conclusions.**

**Confidence:** 2
**Clarity Of Writing:** fair
**Clinical Significance:** good
**Methodological Novelty:** fair
**Overall Rating:** 5

**Experiments And Results:**

fair

**Questions For The Authors:**

I see two different submission for this same paper, please contact the organizers.
Paper ID: 272 and 293 are same.

**Strengths:**

Paper tackles a significant problem (repeatability in clinical AI), proposes a practical solution (minibatches), and supports its claims with a structured experiment tied to a real-world application. The focus on longitudinal consistency addresses a gap in the deployment of neural networks in healthcare, where reliability over time is critical.

**Summary Of The Paper:**

The paper demonstrates that minibatch training mitigates round-off-induced divergence during neural network training, thus ensuring repeatability, a key requirement for AI models in clinical applications.

**Weaknesses:**

I feel like the paper is incomplete, that prevents a full evaluation of its rigor, and the findings' generalizability and depth remain questionable without additional data.=> While a batch size of 2 worked in this case, the paper does not explore larger batch sizes, different architectures, or varied datasets. The solution’s robustness across diverse scenarios is untested here.

=> Paper focuses on one task and one network. While the methods are generalizable, reproducibility across other architectures/tasks isn’t explored.

Paper focuses on a specific task (CNS tumor classification) and hardware setup. Whether the findings apply to more complex models, other clinical applications, or different computational environments is unclear. This narrow scope limits the paper’s broader applicability.

There’s no formal guarantee about when k > 1 is sufficient. But the authors acknowledge this is an empirical insight.

No comparison to alternative solutions (e.g. gradient checkpointing, mixed-precision with stochastic rounding, deterministic GPU kernels).

---

### Official Review · Reviewer_NgsL · 2025-07-20
**Requires more explanation**

**Confidence:** 3
**Clarity Of Writing:** good
**Clinical Significance:** fair
**Methodological Novelty:** fair
**Overall Rating:** 4
**Final Rating:** 6

**Experiments And Results:**

fair

**Questions For The Authors:**

Please see the limitations

**Strengths:**

1. Shows the impact of round off errors
2. Mitigate the issue using mini-batch

**Summary Of The Paper:**

This paper explains how round-off error can cause training repeatability issues bad enough to create difficulties for longitudinal consistency

**Weaknesses:**

1. This is a well-known issue. So, I don't see any novelty.
2. Why this issue is only applicable for clinical models is not clear.
3. How mini-batch solves the issue requires more justification.